# Two Novel Betarhabdovirins Infecting Ornamental Plants and the Peculiar Intracellular Behavior of the Cytorhabdovirus in the Liana *Aristolochia gibertii*

**DOI:** 10.3390/v16030322

**Published:** 2024-02-21

**Authors:** Pedro Luis Ramos-González, Maria Amelia Vaz Alexandre, Matheus Potsclam-Barro, Lígia Maria Lembo Duarte, Gianluca L. Michea Gonzalez, Camila Chabi-Jesus, Alyne F. Ramos, Ricardo Harakava, Harri Lorenzi, Juliana Freitas-Astúa, Elliot Watanabe Kitajima

**Affiliations:** 1Laboratório de Biologia Molecular Aplicada, Centro de Pesquisa e Sanidade Vegetal, Instituto Biológico de São Paulo, Av. Cons. Rodrigues Alves, 1252, São Paulo 04014-002, SP, Brazil; potsclam16@gmail.com (M.P.-B.); gianlucalgonzalez@gmail.com (G.L.M.G.); millachabi@yahoo.com.br (C.C.-J.); ricardo.harakava@sp.gov.br (R.H.); 2Laboratório de Fitovirologia Fisiopatológica, Centro de Pesquisa e Sanidade Vegetal, Instituto Biológico de São Paulo, Av. Cons. Rodrigues Alves, 1252, São Paulo 04014-002, SP, Brazil; maria.alexandre@sp.gov.br (M.A.V.A.); ligia.duarte@sp.gov.br (L.M.L.D.); alyne.ramos@sp.gov.br (A.F.R.); 3Escola Superior de Agricultura Luiz de Queiroz (ESALQ), Universidade de São Paulo, Piracicaba 13418-900, SP, Brazil; ewkitaji@usp.br; 4Instituto Plantarum, Nova Odessa 13380-410, SP, Brazil; hlorenzi@plantarum.com.br; 5Embrapa Mandioca e Fruticultura, Cruz das Almas 44380-000, BA, Brazil; juliana.astua@embrapa.br

**Keywords:** rhabdovirus, betanucleorhabdovirus, high-throughput sequencing, virions, virus particles, cytopathology

## Abstract

Two novel members of the subfamily *Betarhabdovirinae*, family *Rhabdoviridae,* were identified in Brazil. Overall, their genomes have the typical organization 3′-*N-P-P3-M-G-L-5*′ observed in mono-segmented plant-infecting rhabdoviruses. In aristolochia-associated cytorhabdovirus (AaCV), found in the liana aristolochia (*Aristolochia gibertii* Hook), an additional short orphan ORF encoding a transmembrane helix was detected between *P3* and *M*. The AaCV genome and inferred encoded proteins share the highest identity values, consistently < 60%, with their counterparts of the yerba mate chlorosis-associated virus (*Cytorhabdovirus flaviyerbamate*). The second virus, false jalap virus (FaJV), was detected in the herbaceous plant false jalap (*Mirabilis jalapa* L.) and represents together with tomato betanucleorhabdovirus 2, originally found in tomato plants in Slovenia, a tentative new species of the genus *Betanucleorhabdovirus*. FaJV particles accumulate in the perinuclear space, and electron-lucent viroplasms were observed in the nuclei of the infected cells. Notably, distinct from typical rhabdoviruses, most virions of AaCV were observed to be non-enclosed within membrane-bounded cavities. Instead, they were frequently seen in close association with surfaces of mitochondria or peroxisomes. Unlike FaJV, AaCV was successfully graft-transmitted to healthy plants of three species of the genus Aristolochia, while mechanical and seed transmission proved unsuccessful for both viruses. Data suggest that these viruses belong to two new tentative species within the subfamily *Betarhabdovirinae*.

## 1. Introduction

The family *Rhabdoviridae*, a member of the order *Mononegavirales*, comprises virions characterized by bacilliform, bullet-shaped, or rod-shaped morphologies, with most species exhibiting enveloped particles. Invariably, these virions carry single-stranded (ss) negative-sense (-) RNA molecules as their genome [1]. Rhabdoviruses collectively infect invertebrate and vertebrate animals, humans, and plants. They cause diseases that pose significant public health concerns and lead to substantial economic losses [2]. Plant-infecting rhabdoviruses are taxonomically classified into six genera within the subfamily *Betarhabdovirinae* [3]. It should be recalled that members of the former genus “*Nucleorhabdovirus”* were reassigned to the new genera *Alphanucleorhabdovirus*, *Betanucleorhabdovirus*, and *Gammanucleorhabdovirus* [4]. Overall, betarhabdovirins affect gymnosperm and angiosperm plants including crops, ornamentals, and weeds [5,6]. Usually, they are transmitted by arthropods, where they persist and multiply.

Betarhabdovirins of the genus *Cytorhabdovirus* accomplish their replication in the cytoplasm of the infected cells. In contrast, those of the genus *Betanucleorhabdovirus* replicate in the nucleus, and their particles mature by budding into the perinuclear space. Cytorhabdoviruses and betanucleorhabdoviruses possess non-segmented genomes ranging from 10 to 16 kb and consist of partially complementary untranslated regions known as leader and trailer at the 3′- and 5′-ends of the RNA molecules, respectively [7]. In addition to the five genes regularly found in members of the family *Rhabdoviridae*—namely, nucleocapsid protein (N), phosphoprotein (P), matrix protein (M), glycoprotein (G), and polymerase (L)—the genome of betarhabdovirins encodes the protein P3, also called MP (movement protein), required for cell-to-cell movement in plants. Furthermore, their genomes may contain one to several auxiliary genes with functions that are currently unknown. The transmission of several cytorhabdoviruses and betanucleorhabdoviruses occurs through aphids, leafhoppers, planthoppers, or whiteflies [5,8].

Aristolochia (*Aristolochia* spp.) and false jalap (*Mirabilis jalapa* L.) plants are ornamentals that also stand out because of their applications in herbal medicine. Genus Aristolochia encompasses more than 400 species of magnoliid plants of the family Aristolochiaceae. *Aristolochia* spp. are widely distributed in practically all continents and are notable for their unique perianth structures [9]. These plants have a long history of medicinal use in the treatment of abscesses, eczemas, and other long-lasting skin diseases, and as a non-specific immune system stimulant [10]. The genome of *Aristolochia* spp. typifies a hefty pivot in studies of floral biology, developmental genetics, and biochemical pathways, contributing to a better understanding of angiosperm evolution [11]. For their part, four o’clock flower plants, also known as false jalap (family Nyctaginaceae), are native to tropical America, likely from Mexico. They were cultivated by the Aztecs due to their medicinal properties, showy and colorful flowers, and pleasant fragrance. Since then, false jalap plants have been introduced into various continents, becoming the widest-spread ornamental of the genus *Mirabilis*, and sometimes being considered a potentially invasive weed [12]. False jalap plants have a high reproductive rate. They can be easily propagated by seeds and vegetatively using their cuttings and tuberous roots. Therefore, they may be easily dispersed from gardens into nearby areas, which potentially contributes to the spread of the infecting viruses.

In this study, we present the identification and characterization of two betarhabdovirins found infecting false jalap and aristolochia plants, respectively, in Brazil. The infected tissues were examined by transmission electron microscopy to study virion morphologies and cytopathic effects. Phylogenetic and taxonomic descriptions based on the complete genome or full-length coding region of the detected betarhabdovirins are provided. Additionally, transmission assays were conducted using seeds from infected plants, as well as graft or sap extracts, to discern the potential transmission properties of each virus.

## 2. Materials and Methods

### 2.1. Plant Material

Symptomatic leaves of aristolochia (*Aristolochia gibertii* Hook) plants were collected in a botanical garden located at the Instituto Plantarum, in Nova Odessa, State of São Paulo (SP), Brazil (22°46′45.6″ S 47°18′47.6″ W), in 2020. Leaf tissues from symptomatic false jalap (*Mirabilis jalapa* L.) plants were collected in public and private gardens of the city of São Paulo, SP, Brazil, in 2017. Tissues from healthy plants of the studied species were also collected for use as negative controls in molecular tests.

### 2.2. Transmission Electron Microscopy and Molecular Detection of Viruses

Microscopy analyses were carried out at Escola Superior de Agricultura Luiz de Queiroz, Piracicaba, SP, Brazil, whereas the molecular analyses were performed at Instituto Biológico de São Paulo, SP, Brazil. Leaf extracts from symptomatic plants were negatively stained with 1% uranyl acetate in an attempt to observe virions in suspension [13]. For in situ detection of the viral particles, pieces of the infected tissues were pretreated for transmission electron microscopy (TEM) analyses [14]. Copper grids with plant tissue sections embedded in Spurr’s epoxy resin and stained with 3% uranyl acetate and Reynold’s lead citrate were examined under a transmission electron microscope JEOL JEM 1011 (JEOL, Akishima, Japan). Images were recorded digitally.

For molecular analyses, approximately 100 mg of leaf tissues were ground to powder with a mortar and pestle in the presence of liquid nitrogen, and the total RNA was extracted using Trizol Reagent (Life Technologies, Foster City, CA, USA). Five-hundred-nanogram samples of the RNA extracts were used to generate the cDNA solution using random primers following the manufacturer’s recommendations for the GoScriptTM Reverse Transcriptase kit as described by the manufacturer (Promega, Madison, WI, USA). cDNA solutions (3 µL) were used in independent PCR assays for virus detection using the degenerate primer pair RhabF/RhabR [15] and newly designed primer pairs obtained after genomic analyses carried out in this study (Appendix A).

### 2.3. Higth-Throuput Sequencing, Rapid Amplification of cDNA Ends, and In Silico Analysis of Viral Genomes

Small tissue fragments from 10 leaves of infected false jalap and aristolochia plants, comprising approximately 50–100 mg of each one, were independently ground in liquid nitrogen to obtain a powdered sample. Total RNA purification from each sample was performed using TRIzol™ reagent following the manufacturer’s instructions (Life Technologies, Foster City, CA, USA). One hundred nanograms of the RNA extract underwent ribosomal RNA depletion before the high-throughput sequencing (HTS) library preparation. Subsequently, the library was sequenced using HiSeq 2500 Technology (2 × 150 nt paired-end reads) (Illumina, San Diego, CA, USA) at the Animal Biotechnology Laboratory, Escola Superior de Agricultura Luiz de Queiroz, University of São Paulo (Piracicaba, SP, Brazil). The HTS library preparation, sequencing, and quality assessment procedures have previously been described [16]. The obtained reads were processed using SPAdes v3.15.3 [17] and Trinity v2.15.1 [18] de novo assemblers, both implemented on the Galaxy platform 23.2.rc1 (https://usegalaxy.org/, accessed on 31 December 2023) [19].

To identify plant virus-related contigs, BLASTx and BLASTn programs [20] available on the Galaxy platform were employed. Customized databases containing plant viral genomes were obtained from NCBI Virus (https://www.ncbi.nlm.nih.gov/labs/virus/vssi/#/, accessed on 31 December 2023) [21] and utilized for the BLAST analyses. For Aristolochia-associated cytorhabdovirus (AaCV), to confirm the obtained genomic sequence data produced by HTS and to cover the entire viral genome, including the 5′- and 3′-ends, a set of 35 specific primers was designed based on the contig sequence using the Primer3 program [22] (Appendix A). Genomic RNA ends were determined by the rapid amplification of cDNA ends (RACE) using the corresponding primers (Appendix A) and following the manufacturer’s recommendations for the SMARTer RACE 5′/3′ Kit (Clontech Laboratories, Mountain View, CA, USA). All generated amplicons were cloned into the pGEM-T Easy vector (Promega) and subjected to Sanger sequencing with M13 universal primers.

Viral open reading frames (ORFs) were identified using the ORF finder (https://www.ncbi.nlm.nih.gov/orffinder/, accessed on 31 December 2023). The presence of nuclear localization signals (NLS), nuclear export signals (NES), signal peptides (SP), conserved domains, and transmembrane (TM) helices in predicted viral proteins was detected using LocNES (http://prodata.swmed.edu/LocNES/LocNES.php, accessed on 31 December 2023) [23], NLSStradamus (http://www.moseslab.csb.utoronto.ca/NLStradamus/, accessed on 31 December 2023) [24], SignalP v6.0 (https://services.healthtech.dtu.dk/services/SignalP-6.0/, accessed on 31 December 2023) [25], MOTIF Search (https://www.genome.jp/tools/motif/, accessed on 31 December 2023), and TMHMM Server 2.0 (https://services.healthtech.dtu.dk/services/TMHMM-2.0/, accessed on 31 December 2023) [26], respectively. Alignments and identity values of nucleotide and predicted amino acid (aa) sequences were obtained by MAFFT (https://www.ebi.ac.uk/Tools/msa/mafft/, accessed on 31 December 2023) [27]. For phylogenetic analyses, aa sequences of L and G proteins from selected members of the genera *Alphanucleorhabdovirus*, *Betanucleorhabdovirus*, *Gammanucleorhabdovirus*, *Dichorhavirus*, and *Cytorhabdovirus* were retrieved from GenBank (https://www.ncbi.nlm.nih.gov/, accessed on 31 December 2023). In particular, members of the proposed genus “*Gammacytorhabdovirus*” were not considered in the G protein analysis since these viruses lack the ORF *G* [28]. Some viruses of the genus *Varicosavirus* were included as an outgroup in each analysis. Phylogenetically informative regions of the multiple sequence alignments were selected using BMGE v1.12 and Noisy v1.5.12 software [29,30]. The substitution models with the lower BIC scores and the Maximum Likelihood trees were obtained using W-IQ-TREE software v. 1.6.12 (http://iqtree.cibiv.univie.ac.at/, accessed on 31 December 2023) [31]. The trees were edited and visualized using Interactive Tree Of Life (iTOL) v 5 [32].

### 2.4. Viral Transmission Experiments

The plant-to-plant transmission of the tentative betarhabdovirins was evaluated by graft and mechanical means. Attempts to mechanically transmit AaCV included five plants of the following species: *Chenopodium quinoa* Willd., *C. amaranticolor* Coste & Reyn., *Nicotiana tabacum* L. cv. “Turkish” and “TNN”, *N. glutinosa* L., *Datura stramonium* L., *Gomphrena globosa* L., and *Phaseolus vulgaris* L., besides several *Aristolochia* spp. (*A. gibertii*, *A. elegans* Mast., *A. gigantea* Mart. & Zucc., *A. fimbriata* Cham.). Experiments with false jalap virus included ten plants of false jalap and ten of tomato (*Solanum lycopersicum* L. cv. “Santa Cruz”). For mechanical transmissions, extracts were made by macerating symptomatic leaves with phosphate buffer 0.01 M, pH 7, and sodium sulfite 0.1%.

For graft transmission assays, stems from the original symptomatic plants were fitted on seedlings of the same species tested for mechanical transmission. The vertical transmission of both viruses was also tested. Seeds from the original infected plants were collected, and 50 and 30 seeds from aristolochia and false jalap plants, respectively, were sown to verify possible seed transmission of the putative betarhabdovirins. In addition, for AaCV mechanical transmission, buffers containing strong reducing agents (TACM—0.05 M Tris, 0.1% ascorbic acid, 0.1% cysteine, 0.5% 2-mercaptoethanol, adjusted to pH 8; and PDET—phosphate buffer 0.05 M pH 7, 0.005 M DIECA, 0.001 M Na-EDTA, 0.005 M Na-thioglycolate) [33] were assayed. The presence of the viruses in the inoculated plants was assessed by RT-PCR assays using specific primers (Appendix A) and TEM. Qualities of the plant RNA extracts and the cDNA synthesis were assessed by detecting the plant mitochondrial NADH dehydrogenase gene (*nad5*) mRNA as the internal control [33].

## 3. Results and Discussion

### 3.1. Yellowing Symptoms in Vascular Bundles of the Affected Aristolochia and False Jalap Plants

The affected aristolochia plants exhibited a range of symptoms, including yellowish to pale greenish mosaics, and occasionally, mottles. These symptoms were primarily observed along the vascular bundles, occupying the interveinal spaces of the leaves (Figure 1A–C). In false jalap plants, predominant symptoms were observed in midribs, veins, and venules, which showed up yellowish to whitish (Figure 1D,E). In some leaves, the mesophyll next to the veins and venules also displayed yellowish areas. It is important to note that in both plants, the described symptoms cannot be directly attributed to a single infection by the betarhabdovirins identified in this study. In the case of the aristolochia plant, symptoms could be the outcome of mixed infection by the potyvirus bean common mosaic virus and the cucumovirus cucumber mosaic virus, which were also found in that plant, as previously reported [34]. In false jalap, besides the betanucleorhabdovirus, reads corresponding to the genome of a tentative novel emaravirus were also detected by HTS (Alexandre et al., unpublished data).

### 3.2. Rhabdo-like Particles in False Jalap and Aristolochia Infected Cells

Transmission electron microscopy analyses showed the presence of enveloped bacilliform particles 46–80 nm wide and 340–360 nm long in some leaf parenchymal cells of the infected false jalap plants. Viral particles accumulated in the perinuclear space, and nuclear, electron-lucent viroplasms were observed (Figure 2A,B). Both the virion morphology and cellular morphoanatomy resembled those observed in plant cells infected by alphanucleorhabdoviruses and betanucleorhabdoviruses [35,36]. Virus particles of the tentative emaravirus could not be detected.

Negatively stained leaf extracts from symptomatic aristolochia plants also revealed the presence of rhabdovirus-like particles ca. 60–70 nm wide and 300 nm long. These particles were surrounded by a membrane and had an inner rod-shaped structure with periodicities ca. 10 nm apart (Figure 2C). In some instances, naked inner components were also observed (Figure 2D). They appear essentially similar to the described plant rhabdoviruses [35,36]. These naked bacilliform particles, presumed to be rhabdovirus virions, were consistently detected within the cytoplasm of most of the leaf cells (epidermis, mesophyll, vascular region) (Figure 2E–O). The naked particles had an average width of approximately 60 nm and a length of around 300 nm. Interestingly, unlike typical rhabdovirus infections in plant cells, most of these particles were not enclosed within membrane-bound cavities of the endoplasmic reticulum. Instead, they were frequently observed free in the cytoplasm, in association with the surfaces of mitochondria or peroxisomes; often, these organelles formed large aggregates, joined by the viral particles (Figure 2E–I,O). Free particles in the cytoplasm were also found in groups, not associated with mitochondrion or peroxisome (Figure 2N). On a few occasions, complete virions were found in the lumen of the endoplasmic reticulum (Figure 2L,M), and, sporadically, masses of filamentous particles, likely a kind of viroplasm, were observed (Figure 2I,J), as seen with other cytorhabdoviruses [35,36]. Close to the presumptive viroplasm mass, occasionally what was interpreted as rhabdovirus-like particle morphogenesis by a budding process towards the lumen of the endoplasmic reticulum was observed (Figure 2J,K), similar to reports related to known cytorhabdoviruses [35,36].

### 3.3. Genomic Analyses Confirm the Rhabdovirus Nature of the Detected Viruses

To further investigate the potential rhabdoviruses, HTS libraries comprising 18,652,347 and 16,632,531 clean reads were obtained from symptomatic aristolochia and false jalap plants, respectively. Two contigs, one from each library, with approximately 13,000 nucleotides (nt), exhibited a very high-quality match (E-value < 1 × 10^−100^) with sequences of well-known cytorhabdoviruses and betanucleorhabdoviruses, respectively.

Characterization of the cytorhabdovirus genome: BLASTx and BLASTn analyses of the tentative cytorhabdovirus indicated the closest similarities to sequences of yerba mate chlorosis-associated virus (YmCaV, *Cytorhabdovirus flaviyerbamate*, NCBI Reference Sequence (RefSeq) accession number NC055505). The HTS and Sanger-generated sequences, including the RACE ensuing ones, were assembled, resulting in a final contig of 13,128 nt representing the genome of a new cytorhabdovirus, named aristolochia-associated cytorhabdovirus (AaCV, GenBank database accession number of the sequence OR090884).

The analysis of the AaCV complete genome using MAFFT [27] revealed the highest identity values when compared to YmCaV [37], soybean blotchy mosaic virus (SbBMV, GenBank accession number OM681516.1) [38], and cucurbit cytorhabdovirus 1 (CuCV1, *Cytorhabdovirus cucurbitae*, RefSeq NC076898) [39]. These values were consistently below 51%, with the top identity observed with YmCaV (50.27%), and 45.97% and 45.58% in the comparisons with SbBMV and CuCV1, respectively. A total of seven open-reading frames encoding proteins comprising at least 50 aa were identified in the anti-genome strand of AaCV (Figure 3A).

After the 3′-end leader sequence of 160 nt, the identified ORFs included genes *N* (ORF *N*, 1284 nt), *P* (ORF *P*, 1125 nt), *P3* (ORF *P3*, 549 nt), *M* (ORF *M*, 690 nt), *G* (ORF *G*, 1803 nt), and *L* (ORF *L*, 6360 nt), which are orthologous of ORFs found in rhabdoviruses (Figure 3A) (Table 1). An additional short ORF (*P4*), with 174 nt, was identified between *P3* and *M*. Comparisons of the predicted aa sequences from these ORFs, except for P4, showed the highest identity mainly with those of their counterparts in YmCaV, with values consistently lower than 50% aa sequence identity. In these evaluations, the second and third uppermost figures corresponded to comparisons with proteins from SbBMV and CuCV1, but not always in that order.

In most of the intergenic regions (IG) of AaCV, U-reach stretches of conserved sequences were identified (Figure 3A). These sequences likely correspond with the polyadenylation signal at the 3′-end of the preceding ORF. Downstream of those sequences, but less conserved among them, we also detected sequence motifs comprising a short intergenic non-transcribed region and the transcriptional start of the following gene. Remarkably, we failed to detect these sequences in the *P3*–*P4* intergenic region. Whether other sequences can functionally substitute the motif identified in other intergenic regions of the virus remains to be studied. However, it is a feature also observed in other cytorhabdoviruses, and the transcription of *P3* and *P4* in a single messenger RNA has been speculated [41]. At the 5′-end of the AaCV genome, a short trailer sequence of 28 nt long was detected, but complementarity between the leader and trailer sequences was not evident. Transcription initiation or termination signal sequences in the leader and trailer regions of AaCV were not apparent.

Protein N of AaCV, with an estimated molecular weight of 48.16 kDa, presents a predicted nuclear localization signal (NLS) near its COOH-end, between the positions 404 and 423, which is not typically found in cytorhabdoviruses [35]. This protein also exhibits the motif Rhabdo_ncap_2 (PF03216) between the residues 125 and 346. The phosphoprotein (42.15 kDa) possesses the His Kinase A (phospho-acceptor) domain (PF00512) within the positions 218–273 and a nuclear export signal (NES) comprising the aa stretch 294 to 312 (*p* score = 0.87). P3, with 21.43 kDa, has a motif reminiscent of Gemini_BL1 (PF00845, E-value 0.082), which suggests the involvement of this protein in the cell-to-cell movement activity. P4 protein, consisting of 57 residues, does not show any matches with entries deposited in the GenBank, suggesting that ORF *P4* can be classified as an orphan gene [42]. P4 shows a predicted TM domain between positions 4 and 21, which indicates it may act as a viroporin of class I [43]. ORFs encoding viroporin-like proteins or small TM proteins are commonly found in the genome of rhabdoviruses, and their sequence comparison demonstrates the absence of orthologs [44]. In some cytorhabdoviruses, small proteins with predicted TM domains are encoded by auxiliary ORFs, and they have been detected in the nucleus, nuclear membranes, and cell periphery [28,45,46,47]. Protein M has a predicted molecular weight of 26.54 kDa and no remarkable motifs were detected in its sequence. Protein G, 68.49 kDa, has three predicted TM domains, two allocated close to the NH_2_-terminal, comprising the positions 22–39 and 82–101, and the third one near the COOH-terminal of the protein, at positions 564–583. Additionally, the protein G has three predicted N-glycosylation sites. A signal peptide sequence was not detected in this protein. In protein L, typical motifs of RNA polymerases of mononegaviruses were found in the aa positions 226–1069 (Mononeg_RNA_pol, PF00946, E-value 2.5 × 10^−149^) and 1100–1321 (Mononeg_mRNAcap, PF14318, 2.9 × 10^−18^).

Characterization of the betanucleorhabdovirus genome: Bioinformatic analyses of the betanucleorhabdovirus from false jalap plants indicated the presence of the six canonical ORFs of plant rhabdoviruses, i.e., *N*, *P*, *3MP*, *M*, *G*, and *L* [13]. The nucleotide sequence of the full-length coding sequence of the false jalap virus (FaJV, GenBank accession number OQ513467) comprises 13,450 nt and shows the highest identity values with members of the genus *Betanucleorhabdovirus*, subfamily *Betarhabdovirinae* (Figure 3B). Among the definitive species of the genus, maximum nucleotide identity values, <60%, were obtained with datura yellow vein virus (DYVV, *Betanucleorhabdovirus daturae*) and birds-foot trefoil-associated virus (BFTV, *Betanucleorhabdovirus loti*). However, the genome of FaJV showed 85.02% nucleotide sequence identity with that of tomato betanucleorhabdovirus 2 (TBRV2) isolate BOL20AT, a tentative novel betanucleorhabdovirus recently found in tomato plants in Slovenia [48] (Table 2). The second best identity values were with tomato betanucleorhabdovirus 1 (TBRV1), a specimen also identified in infected tomato plants in Slovenia [48].

The nucleocapsid protein of FaJV has 458 residues (~51 kDa) and shows a conserved rhabdovirus nucleoprotein motif (Rhabdo_ncap_2, PF03216) in the aa stretch 46–348 (*p* = 3.4 × 10^−35^). The deduced protein encoded by the ORF *P* is 335 aa in length (37.02 kDa) and presents a weak identity with the PF11678 motif of unknown function. Protein P3, the tentative movement protein according to the BLAST results, has an estimated molecular weight of 36.87 kDa, but no domains, repeats, motifs, or features could be predicted with confidence. The smallest protein encoded by the genome of FaJV is the M protein, with a predicted molecular weight of 30.43 kDa. Protein G has 1881 residues (70.32 kDa), among which the first 17 comprise a signal peptide (*p* = 0.92). This protein also shows four potential N-glycosylation sites in the Asp residues 383, 469, 577, and 585, and a TM domain between residues 553 and 575. The L protein has a predicted molecular weight of 240.14 kDa, and the motives PF00946 (Mononegavirales RNA dependent RNA polymerase) and PF14318 (Mononegavirales mRNA-capping region V) are detected. Extremely conserved sequences are present in the junctions of all detected ORFs (Figure 3B) with a sequencing similar to that observed in other betanucleorhabdoviruses [6]. Variants of this sequence were also detected in the 3′ leader and the 5′-trailer. In addition, the stretches comprising the terminal 20 nts in the 3′ leader and 5′ trailer sequences show a high complementarity.

Phylogenetic analyses using the deduced aa sequences of the L and G proteins of betarhabdovirins revealed similar grouping patterns in branches comprising FaJV and AaCV in both trees (Figure 4A,B). The similar tree topologies described by these two genes in each virus suggest they had an equivalent evolutionary history. AaCV shares a common node with YmCaV, indicating their likely ancestral connection in South America. AaCV and YmCaV have been identified, respectively, in infected plants collected in São Paulo, in the southern region of Brazil, and Misiones Province, in the northeast part of Argentina [37]. FaJV is placed within a group together with sequences of TBRV1 and TBRV2, but this clade has shorter branches than those comprising AaCV.

The close phylogenetic relationship between AaCV and YmCaV, particularly in the G protein, offers valuable insights into the potential vectors responsible for transmitting these viruses among plants. Although there is a lack of experimental data regarding arthropods’ involvement in YmCaV transmission, based on the phylogenetic relationships it has been suggested that *Bemisia tabaci* might have a role in transmitting CuCV1 [39], a virus within the same clade containing YmCaV and AaCV. Furthermore, papaya virus E and bean-associated cytorhabdovirus (BaCV), two viruses belonging to the species *Cytorhabdovirus caricae*, identified in Ecuador and Brazil, respectively, are known to be transmitted by *Bemisia tabaci* [8,49]. Phylogenetically, these viruses form a sister branch to that including YmCaV, CuCV1, and AaCV. 

**Figure 4 viruses-16-00322-f004:**
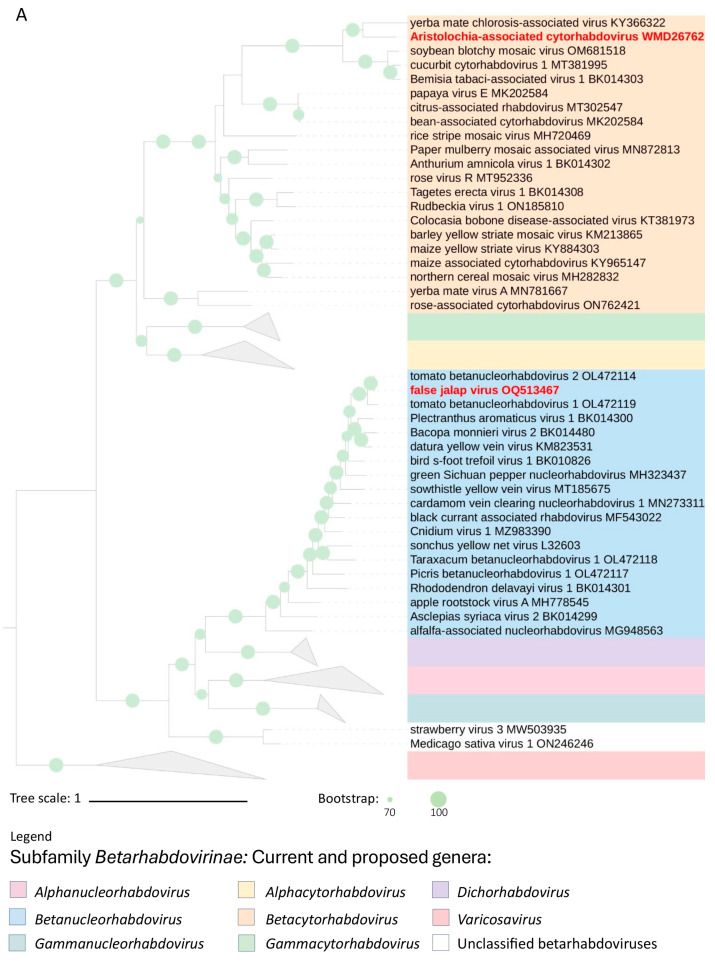
Phylogenetic reconstruction for viruses of the subfamily *Betarhabdovirinae*. (**A**,**B**) Maximum-likelihood phylogenetic trees based on the deduced amino acid sequences of the L and G proteins, respectively. Leaves comprising aristolochia-associated cytorhabdovirus virus and false jalap virus are highlighted in red. Trees were rooted using viruses of the genus *Varicosavirus* as an external group. In (**A**), phylogenetic informative regions of the multiple sequence alignment included 504 residues that were selected using BMGE software [29]. The evolutionary history was inferred based on the model LG + F + R10 [50]. In (**B**), the tree includes 873 residues selected using Noisy software [30], and the evolutionary history was inferred based on the model WAG + F + I + G4 [51]. The bootstrap support values (1000 replications) of branches greater than 70% are indicated next to the corresponding nodes. The scale bar specifies the average number of amino acid substitutions per site. Viruses of the current genus *Cytorhabdovirus* have been assigned to the new genera “*Alphacytorhabdovirus*”, “*Betacytorhabdovirus*”, and “*Gammacytorhabdovirus*”, as recently proposed [28]. Triangles depict collapsed branches.

### 3.4. Under Experimental Conditions, Only AaCV Could Be Graft Transmitted

Non-vector-mediated viral transmission was studied for AaCV and FaJV. Sap extracts from leaves of the symptomatic false jalap plants were inoculated into healthy false jalap and tomato (*Solanum lycopersicum* “Santa Cruz”) plants. In parallel, pieces of diseased branches were grafted onto another set of healthy plants of both species. However, none of those attempts to transmit FaJV were successful. In addition, the virus could not be detected in seedlings germinated from seeds collected from symptomatic plants. The absence of viral infection in each experiment was confirmed by RT-PCR tests using specific primers. Molecular identities among FaJV, TBRV2, TBRV1, and DYVV, and transmission particularities observed in the assays with FaJV, are reminiscent of the description of a tomato-infecting rhabdovirus detected in Southeast Queensland, Australia [52]. Besides apparently similar to DYVV considering the virion morphology and cytopathology of the infected cells, Queensland’s tomato virus was not mechanically transmitted [52,53]. Unfortunately, further molecular or serological traits of the tomato rhabdovirus isolated in Australia remain unknown [52].

Experiments with AaCV demonstrate that this virus is graft-transmitted, although likely at different rates, to *A. gibertii*, *A. fimbriata,* and *A. gigantea* (Table 3), whereas the effort to graft transmit the virus to a plant of *A. elegans* was unsuccessful. Attempts to mechanically transmit AaCV to several species of the genus Aristolochia as well as to a set of indicator plants failed. Vertical transmission by seeds was not observed. The infection of *Aristolochia* species plants to AaCV was confirmed by the detection of amplicons of the expected size, 380 nt, and the presence of rhabdovirus-like particles in sections of leaf tissues. AaCV-infected plants primarily showed mosaic symptoms likely as a consequence of the mixed infection by a potyvirus and a cucumovirus. The sap transmission of AaCV failed, as also verified for the cytorhabdoviruses CuCV1 and BaCV [39,54]. The low efficiency or absence of sap transmission seems to be a common feature of some cytorhabdoviruses [55].

### 3.5. Uncommon Intracellular Behavior of AaCV Is also Detected in Experimentally Infected Aristolochia spp. Plants

The ultrastructural observation of leaf cells of several *Aristolochia* spp. infected by AaCV in graft-transmission experiments also revealed the unique patterns observed in the naturally infected *A. gibertii* plant, where the virus was first detected (Figure 2E–O). In the experimentally or naturally infected plants, presumed virions were detected in all leaf cell types: epidermis, mesophyll parenchyma, and the vascular region. Complete bacilliform particles were found amid aggregates of several mitochondria or peroxisomes, longitudinally attached to the surfaces of those organelles, which could form large aggregates bonded by these particles. This configuration may result from the interaction of receptors on the surface of mitochondria and peroxisomes and the viral surface glycoprotein. The intriguing fact is that virions of AaCV appear free in the cytoplasm, either associated with mitochondria or peroxisomes, but not within membrane-bounded cavities of the endoplasmic reticulum (ER), as described for other cytorhabdoviruses [35,36]. However, it is important to mention that in a few instances, AaCV virions were also found in the lumen of the ER, where they accumulated after morphogenesis by a budding process as described for other cytorhabdoviruses. Given this, it is tempting to speculate that the presence of free virions in the cytoplasm might be related to the small P4 protein encoded by the orphan *P4* ORF, which resembles viroporin of type 1. It may somehow modify ER cisternal membranes containing virions, resulting in the release of virions into the cytoplasm, which may then associate with membranes of mitochondria and peroxisomes. Alternatively, or together with this, P4 may act by modulating the plant response to the cytopathic effect caused by the viral infection. The physiological significance of this process is still unclear.

### 3.6. Two Novel Betarhabdovirins: A New Species of Cytorhabdovirus and a New Divergent Isolate of a Known Betanucleorhabdovirus

Viruses of the genera *Cytorhabdovirus* and *Betanucleorhabdovirus* are assigned to one of their recognized species showing several of the following characteristics: (i) their complete genomes have nucleotide sequence identities of less than 75%; (ii) they occupy different ecological niches, as evidenced by differences in hosts and/or arthropod vectors; and (iii) they can be clearly distinguished in serological tests or by nucleic acid hybridization. In addition, for the description of a new species in the genus *Cytorhabdovirus*, the aa sequence identity in all cognate open reading frames must be less than 80%. Because of the data presented in this study, AaCV must be considered a member of a novel species within the genus *Cytorhabdovirus*, and the binomial name *Cytorhabdovirus aristolochia* is proposed. In the case of FaJV, together with TBRV2, they could be considered virus isolates of the same species, and the binomial name *Betanucleorhabdovirus mirabilis* is suggested.

At present, the subfamily *Betarhabdovirinae* comprises six genera, but new data have provided compelling support for splitting the current genus *Cytorhabdovirus* into the tentative genera: “*Alphacytorhabdovirus*”, “*Betacytorhabdovirus*”, and “*Gammacytorhabdovirus*” [28]. Upon ratification by the ICTV, the proposed reform will lead to the grouping of viruses belonging to the new tentative species “*Cytorhabdovirus aristolochia*” within the genus “*Betacytorhabdovirus*” [28].

## 4. Conclusions

Plants of some species of the genus Aristolochia have been described as hosts of cucumber mosaic virus in Europe and Taiwan, grapevine fanleaf virus in Hungary, and tomato spotted wilt virus in Greece and Hungary [56]. False jalap plants have been detected as the host of Mirabilis mosaic virus (*Caulimovirus tessellomirabilis*) and Mirabilis jalapa mottle virus (*Carlavirus*) in the USA; Parietaria mottle virus (*Ilarvirus*) in Italy; Basella rugose mosaic virus (*Potyvirus*) in China; tomato chlorotic spot virus (*Orthotospovirus*) in Brazil, and two begomoviruses, Mirabilis leaf curl virus and chilli leaf curl India virus, in India [57,58,59,60,61,62,63]. In this work, the range of viruses infecting *Aritolochia gibertii* and *Mirabilis jalapa* plants has been extended through the identification and characterization of two novel viruses, AaCV and FaJV, respectively. In the case of AaCV, its natural potential host range can be broader, also including several related plant species of the genus Aristolochia. AaCV and FaJV are two novel betarhabdovirins that represent a tentative new species of the genus *Cytorhabdovirus* and a divergent isolate of an already known virus of another tentative new species of the genus *Betanucleorhabdovirus*, respectively. The ultrastructure of AaCV-infected aristolochia cells denotes virion distribution and accumulation divergent from those typically produced by cytorhabdoviruses.

## Figures and Tables

**Figure 1 viruses-16-00322-f001:**
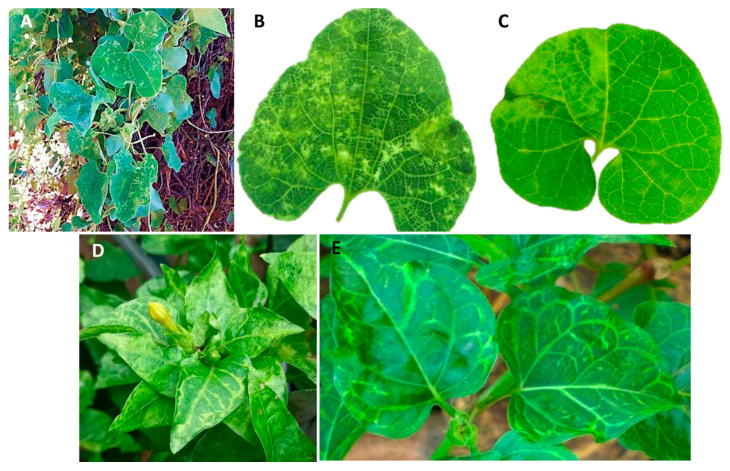
Diseased aristolochia (*Aristolochia gibertii* Hook) and false jalap (*Mirabilis jalapa* L.) plants. (**A**) Aristolochia plants with symptoms of mosaic and mottle in leaves. (**B**,**C**) Detached leaves show contrasting yellowish areas mainly alongside the veins when observed in the backlight. (**D**,**E**) False jalap plants with yellowish to whitish symptoms in midribs, veins, and venules.

**Figure 2 viruses-16-00322-f002:**
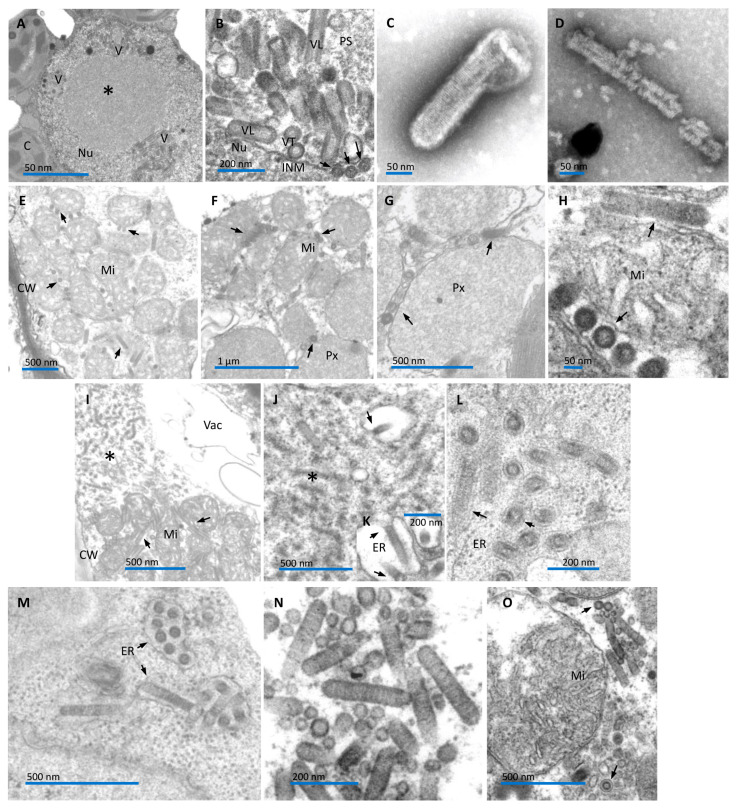
Transmission electron micrographs of ultrathin sections of symptomatic rhabdovirus-infected leaves of *Mirabilis jalapa* (**A**,**B**), *Aristolochia gibertii* plants (**E**–**O**), and leaf dip from diseased *A. gibertii* plants (**C**,**D**). (**A**) Low-magnification image of a spongy parenchyma cell showing pockets of perinuclear cavities containing bacilliform particles (V). A large viroplasm (*) is visible in the center of the nucleus (Nu). C—chloroplast. (**B**) Detail of the presumed rhabdovirus particles within the perinuclear space. A typical bacilliform is evident in longitudinal sections (VL) of the presumed viral particles. The outer membrane and the tubular inner component (nucleocapsid) are depicted in cross-section view (VT). Few naked nucleocapsids (arrows) are visible in the nucleoplasm, close to the inner membrane (INM) of the nuclear envelope, possibly beginning a budding process to complete the morphogenesis. PS—perinuclear space. (**C**,**D**) Membrane-bound and naked particle of a presumed rhabdovirus from *A. gibertii* plants. (**E**) An aggregate of mitochondria (Mi) in a spongy parenchyma cell. Note presumed rhabdovirus-like particles (arrows) between packed mitochondria (Mi). (**F**) In another spongy parenchymal cell, an aggregate of mitochondria (Mi) associated with putative rhabdovirus particles (arrows) is next to particles (arrows)-associated aggregate of peroxisome (Px). (**G**) Detail of rhabdovirus-like particles (arrows)-associated aggregate of peroxisome (Px). (**H**) High-magnification image of the association of presumed rhabdovirus (arrows) with the mitochondrial surface, transversally and longitudinally sectioned. Note that rhabdovirus particles (inner nucleocapsid and surrounding membrane) are free in the cytoplasm, directly in contact with the mitochondrial outer membrane, by their lateral portion. (**I**) Xylem parenchyma cell depicting a mass of filamentous material, interpreted as viroplasm (*), and next to it, rhabdovirus-like particles (arrows), associated with mitochondria (Mi). (**J**) Part of the cytoplasm of a bundle sheath cell, harboring viroplasm (*) showing a rhabdovirus-like particle morphogenesis process by budding (arrow) to the lumen of the endoplasmic reticulum (ER) (arrow). (**K**) Detail of the budding process (arrows) in a spongy parenchymal cell. (**L**,**M**) Presumed rhabdovirus-like particles, contained in the lumen of the ER, in leaf mesophyll parenchymal cells. (**N**) A group of complete rhabdovirus-like particles (inner rod-shaped nucleocapsid and tightly apposed viral membrane) free in the cytoplasm of a mesophyll parenchymal cell. (**O**) In another mesophyll parenchymal cell, a group of rhabdovirus-like particles surrounds a mitochondrion (Mi). Most are free in the cytoplasm (uppermost arrow), but few remain within the lumen of the ER (lowermost arrow).

**Figure 3 viruses-16-00322-f003:**
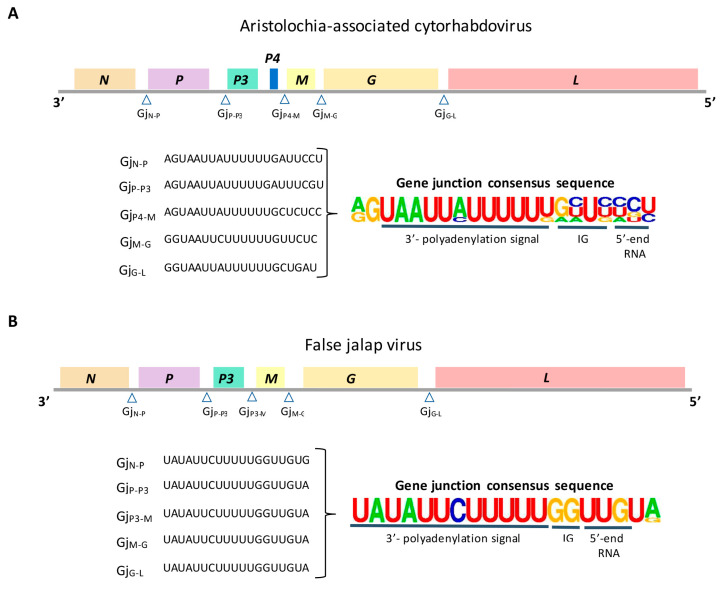
Genomic organization of (**A**) aristolochia-associated cytorhabdovirus (AaCV) and (**B**) false jalap virus. Open reading frames (ORFs) are symbolized by boxes and the open triangles indicate the gene junction (Gj) sequences detected between ORFs. Nucleotide sequences of Gj were aligned, and the consensus sequence, on the right, was generated using WebLogo [40]. IG: intergenic spacer.

**Table 1 viruses-16-00322-t001:** Nucleotide and deduced amino acid identities between AaCV and its closest phylogenetic relationships with members of the genus *Cytorhabdovirus*. The highest values of nt and aa identity in each row are underlined. ORFs are indicated as follows: *N*—nucleocapsid, *P*—phosphoprotein, *P3*—putative movement protein, *P4*—auxiliary gene of unknown function, *M*—matrix protein, *G*—glycoprotein, and *L*—polymerase. YmCaV: yerba mate chlorosis-associated virus. SbBMV: soybean blotchy mosaic virus. CuCV1: cucurbit cytorhabdovirus 1.

Aristolochia-Associated Cytorhabdovirus (GenBank Accession Number OR090884)								
	Genome	ORF						
		*N*	*P*	*P3*	*P4*	*M*	*G*	*L*
Length (nts)	13,128	1284 ^1^	1125	549	174	690	1803	6360
Deduced molecular weight (kDa)	-	48.16	42.15	21.43	6.8	26.54	68.49	246.02
cytorhabdovirus	% of identity (nucleotide/deduced amino acid)							
YmCaV (NC055505)	50.27/-	49.34/36.65	44.47/22.25	49.91/34.46	-	47.22/19.90	48.19/32.45	54.98/49.34
SbBMV (OM681516.1)	45.97/-	43.66/30.64	42.40/19.34	47.38/26.37	-	47.64/20.56	56.41/28.15	52.49/42.31
CuCV1 (NC076898)	45.58/-	44.11/29.37	41.84/18.18	46.65/25.82	-	46.70/20.10	53.51/27.37	52.64/42.44

^1^ Figures indicate the length of the protein-coding region.

**Table 2 viruses-16-00322-t002:** Nucleotide and deduced amino acid identities between FaJV and its closest members or tentative members of the genus *Betanucleorhabdovirus*. The highest values of nt and aa identity in each row are underlined. ORFs are indicated as follows: *N*—nucleocapsid, *P*—phosphoprotein, *P3*—putative movement protein, *M*—matrix protein, *G*—glycoprotein, and *L*—polymerase. TBRV2: tomato betanucleorhabdovirus 2. TBRV1: tomato betanucleorhabdovirus 1. DYVV: datura yellow vein virus. BFTV: birds-foot trefoil-associated virus.

False Jalap Virus (GenBank Accession Number OQ513467)							
	Genome	ORF					
		*N*	*P*	*P3*	*M*	*G*	*L*
Length (nt)	13,450 ^1^	1374 ^3^	1008	978	834	1881	6321
Deduced molecular weight (kDa)	-	51.19	37.02	36.86	30.43	70.32	240.14
betanucleorhabdovirus	% of identity (nucleotide/deduced amino acid)						
TBRV2 ^2^ (OL472116)	85.02/-	89.75/98.25	86.17/88.96	86.12/97.54	86.57/92.81	85.88/94.09	85.39/93.83
TBRV1 ^2^ (OL472119)	66.50/-	72.91/82.49	61.77/53.30	70.86/79.38	67.73/72.64	67.74/70.85	66.72/71.03
DYVV (KM823531)	56.42/-	62.84/59.28	50.43/32.00	52.93/45.49	50.48/38.01	58.02/50.85	58.42/50.05
BFTV (BK010826)	55.54/-	61.23/56.67	48.85/31.00	53.32/39.81	53.43/43.05	54.62/45.54	57.52/50.76

^1^ RACE analyses were not conducted. The composition of the genomic 3′-end leader and 5′ trailer may need further confirmation. ^2^ A tentative member of the genus *Betanucleorhabdovirus*. ^3^ Figures indicate the lengths of the protein-coding regions.

**Table 3 viruses-16-00322-t003:** Graft and mechanically mediated transmission assays of AaCV and FaJV.

Plant Species	Graft Transmission	Mechanical Transmission
	AaCV	
*Aristolochia gibertii* Hook	9/14 ^1^	0/2 ^1^
*A. gigantea* Mart. & Zucc.	8/11	0/2
*A. fimbriata* Cham. & Schltdl	2/7	0/2
*A. elegans* D. Parodi	0/1	-
*Chenopodium quinoa* Willd	-	0/5
*C. amaranticolor* Coste & Reyn.	-	0/5
*Nicotiana tabacum* L. cv. “TNN”	-	0/5
*N. tabacum* L. cv. “Turkish”	-	0/5
*N. glutinosa* L.	-	0/5
*Datura stramonium* L.	-	0/5
*Gomphrena globosa* L.	-	0/5
*Phaseolus vulgaris* L.	-	0/5
	FaJV	
*Solanum lycopersicum* L.	0/6	0/10
*Mirabilis jalapa* L.	0/6	0/10

^1^ Number of infected plants detected by RT-PCR/total of inoculated plants.

## Data Availability

Genomic sequences of viruses described in this study are available at the GenBank database under accession numbers OQ513467 (false jalap virus), and OR090884 (Aristolochia-associated cytorhabdovirus).

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
