# Peer review of "Two Novel Betarhabdovirins Infecting Ornamental Plants and the Peculiar Intracellular Behavior of the Cytorhabdovirus in the Liana Aristolochia gibertii"

_viruses, 2024, doi:10.3390/v16030322_

Round 1
Reviewer 1 Report
Comments and Suggestions for Authors
Ramos-González et al. report the identification of novel negative-stranded RNA viruses (family Rhabdoviridae) of ornamental plants. The complete viral genomic sequences were determined using high-throughput sequencing and rapid amplification of cDNA ends. The findings are nicely supported by viral particles imaging in infected plant samples and plant-to-plant transmission assays.
The study is interesting and presents valuable information.
I have just a couple of comments:
- Rhabdoviruses present (partial) terminal genome complementarity (e.g. Arch Virol. 2021 Jun;166(6):1615-1622. doi: 10.1007/s00705-021-05040-y.). I wonder if 5' trailer/3' leader complementarity was conserved and if the authors can display this information for instance in fig. 3.
- Replace "MAFTT" with "MAFFT" (l.152 and 270)
Author Response
São Paulo, February 17th, 2024
To Ms Nova Peng
Assistant Editor
MDPI (Wuhan).
To Reviewers,
Dear Drs.
The authors would like to thank your cooperative and helpful suggestions to improve the quality and comprehensibility of our ms. Below we list our commentaries to your suggestions and questions. As requested, changes and new data were introduced in the ms—word file (.doc) with tracked changes.
The authors.
Comments and answers to the reviewers
Reviewer 1.
Ramos-González et al. report the identification of novel negative-stranded RNA viruses (family Rhabdoviridae) of ornamental plants. The complete viral genomic sequences were determined using high-throughput sequencing and rapid amplification of cDNA ends. The findings are nicely supported by viral particles imaging in infected plant samples and plant-to-plant transmission assays.
The study is interesting and presents valuable information.
I have just a couple of comments:
- Rhabdoviruses present (partial) terminal genome complementarity (e.g. Arch Virol. 2021 Jun;166(6):1615-1622. doi: 10.1007/s00705-021-05040-y.). I wonder if 5' trailer/3' leader complementarity was conserved and if the authors can display this information for instance in fig. 3.
Authors: Done. The information was added in the section where the genomes of each virus are described. See lines 363-364 and 301-302.
- Replace "MAFTT" with "MAFFT" (l.152 and 270):
Authors: Done
Reviewer 2
The authors report on two plant rhabdoviruses infecting ornamental plants in Brazil and discuss their intracellular behavior in host cells. One virus is thought to be a member of the novel species of the genus Cytorhabdovirus, and the other is a strain of a known betabuchreorahbovirus species within the subfamily Betarhabdovirinae.The experiment is well designed and the results appear to be scientifically sound, and the manuscript is generally well written, including scientific terminology. This paper is considered worthy of publication in Viruses, but there are some minor issues that need to be addressed before it can be accepted.
Comments
1.The title and manuscript refer to "Betarhabdovirids", is this an appropriate description to associate the virus with the subfamily Betarhabdovirinae rather than the family Rhabdoviridae? , Also, “Cytorhabdovirus” should be italicized in the title.
Authors: The term betarhabdovirids has been replaced with the term betarhabdovirins, which is, indeed, the correct form to call members of the subfamily Betarhabdovirinae. The suffix -virids is used for families, whereas the suffix -virins is used for subfamilies. Hence it should be “betarhabdovirins” because the taxon is Betarhabdovirinae. This has been modified across the ms. On the other hand, the term cytorhabdovirus in the title should not be italicized since it indicates a member of the genus and not the genus Cytorhabdovirus by itself.
- Historically, plant rhabdoviruses have been divided into two groups of viruses, assumed to be nuclear and cytoplasmic types (Nucleorhabdovirus and Cytorhabdovirus), and have been split into several existing genera. Since this study will also report the discovery of both types of plant rhabdoviruses, it is nice to introduce this information as background. A reorganization of the genus Cytorhabdovirus is proposed as the authors mentioned in the R&D section, so it is also helpful to include this in the Introduction section.
Authors: Done. The information concerning the former genus Nucleorhabdovirus was included in the Introduction section. See lines 46-48. On the other hand, the subdivision of the current genus Cytorhabdovirus has not been ratified by the ICTV yet, thus this information has been only presented as part of the discussion section.
- A tentative novel emaravirus was also detected in false jalap (L197-198). Therefore, it would be good to have comments on whether potential mature virions corresponding to this new virus were identified during leaf dip or intracellular observations.
Authors: The sequence of the tentative novel emaravirus was identified by NGS. However, viral particles were not visualized by electron microscopy either in leaf-dip or ultrathin sections analyses. New experiments are on course to further characterize this tentative new virus. The information was added in lines 205-206.
- In the text (L278−280) and tables (Tables 1 and 2), each gene is indicated by its ORF length. However, since the actual gene is the mRNA coding region, it is recommended to indicate that it is the length of the protein coding region or to add the estimated gene length.
Authors: The phrase “Figures indicate the length of the protein-coding region” has been added in the footnote of Tables 1 and 2.
- Since the known gammacytorhadboviruses appear to be deficient in the G gene (Bejerman et al., 2023), it would be better to mention this; a reminder that the G phylogenetic tree does not include as a group in Fig 4B.
Authors: A phrase indicating this fact has been added in the M&M section. Lines 150-151. “In particular, gammacytorhabdoviruses were not considered in the G protein analysis since these viruses lack the ORF G [27].”
- The viroporin-like gene (small P4 protein) function is an interesting discussion, but they could add aductive references to other viral examples for its function (L313-315). Also, whether similar intracellular findings have been obtained in other cytorahbdoviruses encoding viroporin-like genes could further this discussion.
Authors: Done. Novel references have been added. They include studies of cellular protein localization from which the virus protein functions may be inferred.
- The authors have successfully grafted to transfer AaCV on Aristolochia spp. and would like to know if disease symptoms occur on successfully propagated scions.
Authors: In the aristolochia plants where the cytorhabdovirus was successfully graft-transmitted, mosaic symptoms were observed, most likely because of the presence of a mixed infection AaCV-potyvirus-cucumovirus. See lines 431-432.
L35 In keywords,“virons or virus particles” is better than “virion particles”.
Authors: Done.
L42 as “their” genome
Authors: Done.
L52 mature “by” budding
Authors: Done.
L57 MP “(movement protein)”
Authors: Done.
L177, 0.05 M
Authors: Done
L178, Tris 0.1%
Authors: Done
L204, “Monopartite“ could be removed?
Authors: Done
L234, L241, “A.” gibertii
Authors: Done
L242, L243, Mitochondria are indicated as “Mi” in the panels.Please include an explanation of the cell wall abbreviation “CW”
Authors: Done
L270, MAF”F”T
Authors: Done
L272, The accession No. (OM681516.1) will be consistent with Table 1 whether to include up to version.
Authors: Done
L287, do the viruses have similar transcription initiation or termination signal sequences in the leader and trailer regions of the genomes (Fig. 3)?
Authors: This is available in lines 365-366 for the betanucleorhabdovirus. We decided not to include the sequences in Figure 3 because RACE analysis has not been carried out. In the case of the cytorhabdovirus, these sequences were not apparent. This has been clearly explained in lines 302-304.
L320, NH”2”, half-width subscript
Authors: Done
L321, COOH-terminal
Authors: Done

Reviewer 2 Report
Comments and Suggestions for Authors
The authors report on two plant rhabdoviruses infecting ornamental plants in Brazil and discuss their intracellular behavior in host cells. One virus is thought to be a member of the novel species of the genus Cytorhabdovirus, and the other is a strain of a known betabuchreorahbovirus species within the subfamily Betarhabdovirinae.The experiment is well designed and the results appear to be scientifically sound, and the manuscript is generally well written, including scientific terminology. This paper is considered worthy of publication in Viruses, but there are some minor issues that need to be addressed before it can be accepted.
Comments
1.The title and manuscript refer to "Betarhabdovirids", is this an appropriate description to associate the virus with the subfamily Betarhabdovirinae rather than the family Rhabdoviridae? Also, “Cytorhabdovirus” should be italicized in the title.
2. Historically, plant rhabdoviruses have been divided into two groups of viruses, assumed to be nuclear and cytoplasmic types (Nucleorhabdovirus and Cytorhabdovirus), and have been split into several existing genera. Since this study will also report the discovery of both types of plant rhabdoviruses, it is nice to introduce this information as background. A reorganization of the genus Cytorhabdovirus is proposed as the authors mentioned in the R&D section, so it is also helpful to include this in the Introduction section.
3. A tentative novel emaravirus was also detected in false jalap (L197-198). Therefore, it would be good to have comments on whether potential mature virions corresponding to this new virus were identified during leaf dip or intracellular observations.
4. In the text (L278−280) and tables (Tables 1and 2), each gene is indicated by its ORF length. However, since the actual gene is the mRNA coding region, it is recommended to indicate that it is the length of the protein coding region or to add the estimated gene length.
5. Since the known gammacytorhadboviruses appear to be deficient in the G gene (Bejerman et al., 2023), it would be better to mention this; a reminder that the G phylogenetic tree does not include as a group in Fig 4B.
6. The viroporin-like gene (small P4 protein) function is an interesting discussion, but they could add aductive references to other viral examples for its function (L313-315). Also, whether similar intracellular findings have been obtained in other cytorahbdoviruses encoding viroporin-like genes could further this discussion.
7. The authors have successfully grafted to transfer AaCV on Aristolochia spp. and would like to know if disease symptoms occur on successfully propagated scions.
Some others
L35 In keywords,“virons or virus particles” is better than “virion particles”.
L42 as “their” genome
L52 mature “by” budding
L57 MP “(movement protein)”
L177, 0.05 M
L178, Tris 0.1%
L204, “Monopartite“ could be removed?
L234, L241, “A.” gibertii
L242, L243, Mitochondria are indicated as “Mi” in the panels.Please include an explanation of the cell wall abbreviation “CW”
L270, MAF”F”T
L272, The accession No. (OM681516.1) will be consistent with Table 1 whether to include up to version.
L287, do the viruses have similar transcription initiation or termination signal sequences in the leader and trailer regions of the genomes (Fig. 3)?
L320, NH”2”, half-width subscript
L321, COOH-terminal
Author Response
São Paulo, February 17th, 2024
To Ms Nova Peng
Assistant Editor
MDPI (Wuhan).
To Reviewers,
Dear Drs.
The authors would like to thank your cooperative and helpful suggestions to improve the quality and comprehensibility of our ms. Below we list our commentaries to your suggestions and questions. As requested, changes and new data were introduced in the ms. word file (.doc) with tracked changes.
The authors.
Comments and answers to the reviewers
Reviewer 1.
Ramos-González et al. report the identification of novel negative-stranded RNA viruses (family Rhabdoviridae) of ornamental plants. The complete viral genomic sequences were determined using high-throughput sequencing and rapid amplification of cDNA ends. The findings are nicely supported by viral particles imaging in infected plant samples and plant-to-plant transmission assays.
The study is interesting and presents valuable information.
I have just a couple of comments:
- Rhabdoviruses present (partial) terminal genome complementarity (e.g. Arch Virol. 2021 Jun;166(6):1615-1622. doi: 10.1007/s00705-021-05040-y.). I wonder if 5' trailer/3' leader complementarity was conserved and if the authors can display this information for instance in fig. 3.
Authors: Done. The information was added in the section where the genomes of each virus are described. See lines 363-364 and 301-302.
- Replace "MAFTT" with "MAFFT" (l.152 and 270):
Authors: Done
Reviewer 2
The authors report on two plant rhabdoviruses infecting ornamental plants in Brazil and discuss their intracellular behavior in host cells. One virus is thought to be a member of the novel species of the genus Cytorhabdovirus, and the other is a strain of a known betabuchreorahbovirus species within the subfamily Betarhabdovirinae.The experiment is well designed and the results appear to be scientifically sound, and the manuscript is generally well written, including scientific terminology. This paper is considered worthy of publication in Viruses, but there are some minor issues that need to be addressed before it can be accepted.
Comments
1.The title and manuscript refer to "Betarhabdovirids", is this an appropriate description to associate the virus with the subfamily Betarhabdovirinae rather than the family Rhabdoviridae? , Also, “Cytorhabdovirus” should be italicized in the title.
Authors: The term betarhabdovirids has been replaced with the term betarhabdovirins, which is, indeed, the correct form to call members of the subfamily Betarhabdovirinae. The suffix -virids is used for families, whereas the suffix -virins is used for subfamilies. Hence it should be “betarhabdovirins” because the taxon is Betarhabdovirinae. This has been modified across the ms. On the other hand, the term cytorhabdovirus in the title should not be italicized since it indicates a member of the genus and not the genus Cytorhabdovirus by itself.
- Historically, plant rhabdoviruses have been divided into two groups of viruses, assumed to be nuclear and cytoplasmic types (Nucleorhabdovirus and Cytorhabdovirus), and have been split into several existing genera. Since this study will also report the discovery of both types of plant rhabdoviruses, it is nice to introduce this information as background. A reorganization of the genus Cytorhabdovirus is proposed as the authors mentioned in the R&D section, so it is also helpful to include this in the Introduction section.
Authors: Done. The information concerning the former genus Nucleorhabdovirus was included in the Introduction section. See lines 46-48. On the other hand, the subdivision of the current genus Cytorhabdovirus has not been ratified by the ICTV yet, thus this information has been only presented as part of the discussion section.
- A tentative novel emaravirus was also detected in false jalap (L197-198). Therefore, it would be good to have comments on whether potential mature virions corresponding to this new virus were identified during leaf dip or intracellular observations.
Authors: The sequence of the tentative novel emaravirus was identified by NGS. However, viral particles were not visualized by electron microscopy either in leaf-dip or ultrathin sections analyses. New experiments are on course to further characterize this tentative new virus. The information was added in lines 205-206.
- In the text (L278−280) and tables (Tables 1 and 2), each gene is indicated by its ORF length. However, since the actual gene is the mRNA coding region, it is recommended to indicate that it is the length of the protein coding region or to add the estimated gene length.
Authors: The phrase “Figures indicate the length of the protein-coding region” has been added in the footnote of Tables 1 and 2.
- Since the known gammacytorhadboviruses appear to be deficient in the G gene (Bejerman et al., 2023), it would be better to mention this; a reminder that the G phylogenetic tree does not include as a group in Fig 4B.
Authors: A phrase indicating this fact has been added in the M&M section. Lines 150-151. “In particular, gammacytorhabdoviruses were not considered in the G protein analysis since these viruses lack the ORF G [27].”
- The viroporin-like gene (small P4 protein) function is an interesting discussion, but they could add aductive references to other viral examples for its function (L313-315). Also, whether similar intracellular findings have been obtained in other cytorahbdoviruses encoding viroporin-like genes could further this discussion.
Authors: Done. Novel references have been added. They include studies of cellular protein localization from which the virus protein functions may be inferred.
- The authors have successfully grafted to transfer AaCV on Aristolochia spp. and would like to know if disease symptoms occur on successfully propagated scions.
Authors: In the aristolochia plants where the cytorhabdovirus was successfully graft-transmitted, mosaic symptoms were observed, most likely because of the presence of a mixed infection AaCV-potyvirus-cucumovirus. See lines 431-432.
L35 In keywords,“virons or virus particles” is better than “virion particles”.
Authors: Done.
L42 as “their” genome
Authors: Done.
L52 mature “by” budding
Authors: Done.
L57 MP “(movement protein)”
Authors: Done.
L177, 0.05 M
Authors: Done
L178, Tris 0.1%
Authors: Done
L204, “Monopartite“ could be removed?
Authors: Done
L234, L241, “A.” gibertii
Authors: Done
L242, L243, Mitochondria are indicated as “Mi” in the panels.Please include an explanation of the cell wall abbreviation “CW”
Authors: Done
L270, MAF”F”T
Authors: Done
L272, The accession No. (OM681516.1) will be consistent with Table 1 whether to include up to version.
Authors: Done
L287, do the viruses have similar transcription initiation or termination signal sequences in the leader and trailer regions of the genomes (Fig. 3)?
Authors: This is available in lines 365-366 for the betanucleorhabdovirus. We decided not to include the sequences in Figure 3 because RACE analysis has not been carried out. In the case of the cytorhabdovirus, these sequences were not apparent. This has been clearly explained in lines 302-304.
L320, NH”2”, half-width subscript
Authors: Done
L321, COOH-terminal
Authors: Done
